# Deep Neural Networks for Quantitative Damage Evaluation of Building Losses Using Aerial Oblique Images: Case Study on the Great Wall (China)

Yiping Gong [1], Fan Zhang [1], Xiangyang Jia [1], Xianfeng Huang [1,2,*], Deren Li [1] and Zhu Mao [1]

1  The State Key Laboratory of Information Engineering in Surveying, Mapping and Remote Sensing, Wuhan University, 129 Luoyu Road, Wuhan 430079, China; gongyp15@163.com (Y.G.); zhangfan@whu.edu.cn (F.Z.); jiaxiangyang@whu.edu.cn (X.J.); drli@whu.edu.cn (D.L.); maoz@whu.edu.cn (Z.M.)
2  Institute of Yangtze River Civilization and Archaeology, Wuhan University, 129 Luoyu Road, Wuhan 430079, China
*  Correspondence: hwangxf@gmail.com

**Abstract:** Automated damage evaluation is of great importance in the maintenance and preservation of heritage structures. Damage investigation of large cultural buildings is time-consuming and labor-intensive, meaning that many buildings are not repaired in a timely manner. Additionally, some buildings in harsh environments are impossible to reach, increasing the difficulty of damage investigation. Oblique images facilitate damage detection in large buildings, yet quantitative damage information, such as area or volume, is difficult to generate. In this paper, we propose a method for quantitative damage evaluation of large heritage buildings in wild areas with repetitive structures based on drone images. Unlike existing methods that focus on building surfaces, we study the damage of building components and extract hidden linear symmetry information, which is useful for localizing missing parts in architectural restoration. First, we reconstruct a 3D mesh model based on the photogrammetric method using high-resolution oblique images captured by drone. Second, we extract 3D objects by applying advanced deep learning methods to the images and projecting the 2D object segmentation results to 3D mesh models. For accurate 2D object extraction, we propose an edge-enhanced method to improve the segmentation accuracy of object edges. 3D object fragments from multiple views are integrated to build complete individual objects according to the geometric features. Third, the damage condition of objects is estimated in 3D space by calculating the volume reduction. To obtain the damage condition of an entire building, we define the damage degree in three levels: no or slight damage, moderate damage and severe damage, and then collect statistics on the number of damaged objects at each level. Finally, through an analysis of the building structure, we extract the linear symmetry surface from the remaining damaged objects and use the symmetry surface to localize the positions of missing objects. This procedure was tested and validated in a case study (the Jiankou Great Wall in China). The experimental results show that in terms of segmentation accuracy, our method obtains results of 93.23% mAP and 84.21% mIoU on oblique images and 72.45% mIoU on the 3D mesh model. Moreover, the proposed method shows effectiveness in performing damage assessment of objects and missing part localization.

**Keywords:** quantitative damage evaluation; loss of material; buildings with linear repetitive symmetrical structure; symmetry surface extraction; missing object localization; aerial oblique image

## 1. Introduction

Cultural heritage around the world, such as ancient buildings, grottoes, and murals, is suffering destruction and degradation due to natural disasters, anthropogenic activities, and a lack of maintenance. Damage investigation of large cultural buildings, such as onsite surveys conducted with the help of instruments [1,2], is time-consuming, labor-intensive, and technically complex, and it cannot satisfy the needs of timely maintenance

and preservation of large heritage structures [3]. In addition, some cultural buildings in remote, wild areas with harsh environments are impossible to reach and perform spot investigations of. In recent years, vision technology has come to be used more widely in the field of structural health monitoring. A structural health monitoring (SHM) system [4,5] is adopted to remotely monitor the conditions of historical buildings, evaluate data through computer-based protocols and define proper maintenance strategies. However, the overall cost of a monitoring system is high when monitoring large scenes. Thus, automated non-contact damage detection and evaluation are of great importance. With the development of remote sensing technology, different types of datasets, such as optical images [6–8], SAR [9], and LIDAR [10], are widely used for damage detection, greatly accelerating the process of damage detection in large heritage buildings. However, damage detection using satellite images or LIDAR data as the only input has certain limitations:

1.  For optical images, it is difficult to distinguish dense objects and extract quantitative damage information, such as height and volume. Moreover, the recognition of objects from images may be influenced by surrounding objects, such as vegetation.
2.  For LIDAR data, which are restricted to the shooting perspective, both ground-based and airborne laser scanners fail to collect all views of object data and thus cannot perform comprehensive damage analysis. In addition, the data acquisition cost is very high.

Oblique images offer a reliable oblique perspective without cloud coverage effects, providing detailed information on both the facades and roofs of buildings [11–13]. Multi-view images can mitigate the problem of inaccurate object recognition caused by occlusion. Moreover, with large block overlap, they can be used to produce 3D models utilizing stereoscopic methods, which provide the essential geometric features for the quantitative evaluation of buildings. With the benefit of large coverage, low cost, and fast data acquisition, oblique images have been recognized as the most suitable data source to provide timely data for automated detection of damage in larger areas [12,14–16].

Recently, some efforts have promoted the use of oblique images and derived 3D models in damage detection for historical buildings [12,14,15,17–19]. These methods primarily focus on the detection of visible damage on building surfaces, including identification of the damage type and localization of the damage region. However, for structural damage, such as destroyed roofs, collapsed walls, and other damaged components, these methods do not carry out further quantitative analysis of the damage, such as the loss of material, which is useful information in the architectural restoration process. Therefore, in this paper, we study quantitative damage assessment of a building from the perspective of material loss, including damage to a single object and missing objects in the building.

The case study is part of the Great Wall, called Jiankou Great Wall, in Beijing in northern China. The Great Wall is a typical Chinese human-made building composed of repetitive objects called Great Wall merlons. These repetitive merlons are generally stacked in a specific manner. Jiankou Great Wall is the most precarious section of the Great Wall in Beijing and is naturally severely weathered. Due to the steep terrain, many walls are severely damaged and cannot be repaired quickly. Therefore, in this paper, we propose a remote non-contact method for quantitative damage evaluation of large structures using high-resolution aerial oblique images. The procedure consists of four stages: 3D mesh model reconstruction, 3D object segmentation, damage assessment corresponding to the loss of material, and symmetric surface extraction and missing object localization. The main contributions of this paper are threefold:

1.  Quantitative damage evaluation of repetitive objects: In contrast to most studies, which focus on damage to building surfaces, we analyze the damage condition of the repetitive objects that compose the building. Repetitive objects with common properties, such as shape, area, and volume, make automatic damage estimation possible. To obtain the damage condition of each object, we collect statistics on the volume reduction information, based on which a damage degree is generated.

2.  Symmetry structure extraction and missing object localization: Chinese buildings are usually composed of repetitive objects in a symmetrical form. Therefore, for buildings with repetitive and symmetrical structures, we extract the symmetry surface of the building based on the spatial distribution of the remaining objects and then use the symmetry information to localize missing objects.

3.  Edge-enhanced convolutional neural network (CNN) for accurate object segmentation: Quantitative damage evaluation of objects requires an accurate segmentation of objects. As it is difficult to distinguish dense or connected objects directly from mesh models, we transform 3D object segmentation to 2D object segmentation by taking advantage of state-of-the-art CNNs. To extract high-quality objects, we propose an edge-enhanced method to improve the segmentation accuracy at the object edges.

The remainder of this paper is organized as follows. In Section 2, related work, including damage detection, object segmentation, and damage evaluation methods, is reviewed. In Section 3, the proposed framework is carefully illustrated. Experiments and analysis are presented in Section 4. The discussion and conclusion are given in Sections 5 and 6, respectively.

## 2. Related Work

### 2.1. Damage Detection

Most damage detection methods [14,17,19] extract defect types and regions in an image based on object-based image analysis (OBIA). However, these methods localize only damaged areas, without quantitative damage analysis. With the development of digital photogrammetry, some works [15,18] have used geometric characteristics derived from 3D point clouds or surface models along with image-derived features for damage detection. Vetrivel et al. (2015) [17] detected gaps due to damage to fully or partially intact buildings through the combined use of images and 3D models. Fernandez-Galarreta et al. (2015) [12] extracted severely damaged buildings directly from derived 3D point clouds. Benefiting from the advance of deep learning, Vetrivel et al. (2018) [15] used CNN features extracted from UAV images and geometric characteristics derived from 3D point clouds, both independently and in combination, to detect damage in buildings. However, these methods fail to generate quantitative statistics on building damage. Therefore, the main purpose of this research is to develop a possible approach for the quantitative evaluation of missing building parts.

### 2.2. 2D Object Segmentation and Edge Enhancement

Object segmentation requires the correct detection of objects as well as precise segmentation of each object. It therefore combines two tasks: object detection and semantic segmentation. Object detection aims to classify all objects of interest in an image and localize each using a bounding box. Semantic segmentation focuses on the pixel-level multiclass classification of the whole image. Thus, inspired by powerful region-based CNNs such as R-CNN [20], fast/faster R-CNN [21,22], and numerous extensions [23–26], some works [27,28] start from object detection and then perform semantic segmentation within each object bounding box. On the other hand, driven by the effectiveness of semantic segmentation, such as fully convolutional networks (FCNs) [29], U-Net [30], and DeepLab [31], some works [32–37] start from pixel-level or region-level segmentation and then group pixels to form different instances according to their object classes.

Although tremendous progress has been made in the field of instance segmentation, the problem of inaccurate segmentation still exists. Inaccurate segmentation mainly occurs at object edges because the edges of objects are usually surrounded by more complex and confusing backgrounds than internal regions. In addition, the number of edge pixels is much less than the number of internal pixels, which means that the segmentation model is mostly dominated by the easy internal pixels and does not perform well at edge pixels.

To address this problem, edge detection methods [38,39] and improved loss functions [40–42] have been proposed to overcome poor segmentation at the edges. The former adds a branch for edge detection in parallel with the existing branch for object mask predic-

tion and uses edge prediction to strengthen the coarse segmentation results. This additional learning task increases the computational cost. The latter forces the model to learn edge pixels by upweighting their contributions to loss functions during the training process. In this work, we apply the second strategy and use the gradient to strengthen the learning of edges.

### 2.3. Damage Evaluation and Rating

The evaluation criteria are different when evaluating different types of damage. For crack damage, the widely used criteria include length, orientation, and width. Zhu et al. (2011) [43] retrieved concrete crack properties, such as length, orientation, and width, for automated post-earthquake structural condition assessment. Unlike previous studies of finding cracks in paintings, Jahanshahi et al. (2013) [44] used 3D depth perception of a scene to adaptively detect and quantify cracks after extracting pixel-level crack segmentation in the images. For building-level damage, such as change detection in buildings, the commonly used indicator is height [45]. For degradation damage, the widely used criterion is volume; therefore, we quantitatively assess the damage of objects based on volume reduction.

For the damage rating of masonry and reinforced buildings, reference has been made in the European Macroseismic Scale 1998 (EMS98) [46], which includes five damage grades: slight damage, moderate damage, heavy damage, very heavy damage, and destruction. Inspired by the damage categories of EMS98, for the degraded damage rating, we simply define the damage grades in three levels based on the reduction in volume: no or slight damage (less than 30%), moderate damage (30~60%), and severe damage (more than 60%). For heavily destroyed objects that are collapsed or missing, we regrade their damage degrees as destruction and localize their positions based on the analysis of the building structure.

### 3. The Method

For objects that are not isolated but attached to each other, it is difficult to distinguish them directly on 3D models. Therefore, we perform object segmentation from images and project them to 3D mesh models. Based on the 3D segmentation results, an analysis of the volume and building structure is conducted to obtain the damage condition of the objects and the positions of missing objects. The overall architecture of our proposed method is presented in Figure 1. The details of each step are as follows.

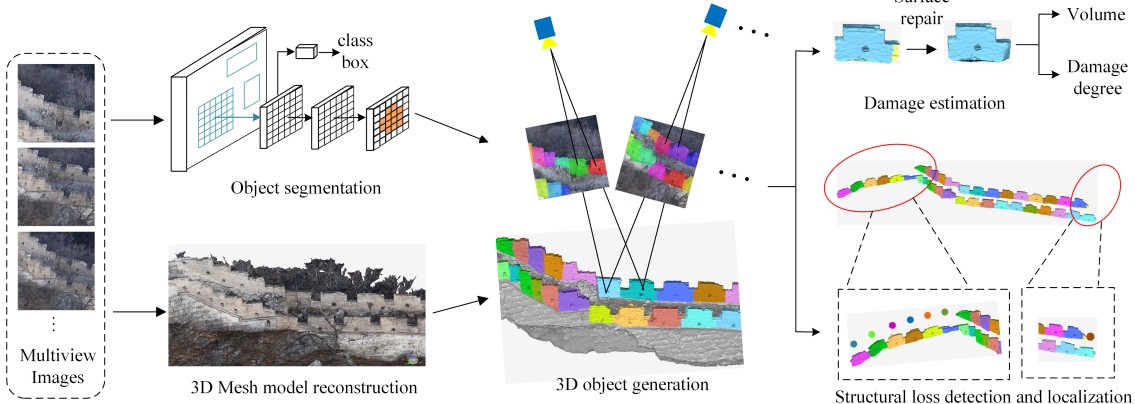

**Figure 1.** Pipeline for quantitative damage evaluation of buildings with a linear repetitive symmetry structure.

(1) 3D mesh model reconstruction. We collect sequences of high-resolution oblique images with a drone and reconstruct the 3D mesh model based on the photogrammetric method.

(2) 3D object segmentation. We extract 3D objects by applying the advanced deep learning method to the images and projecting 2D object segmentation results to the 3D mesh models. First, to segment accurate 2D objects, we propose an edge-enhanced method

to improve the segmentation accuracy of objects, especially at the edges. Second, the segmentation results are projected to the aforementioned mesh model to obtain 3D object fragments. Then, visible outliers coming from a neighboring object or the surrounding area are eliminated according to the characteristics of connectivity. Finally, the 3D object fragments from different viewing directions are integrated to build complete 3D objects according to the geometric features.

(3) Damage assessment corresponding to the loss of material. On the basis that the same objects have the same properties, the damage condition of 3D objects is estimated based on the volume reduction. Because the surfaces of 3D objects extracted from the mesh model are not closed, we first seal their surfaces before performing volume calculations. To obtain the damage condition of the entire building, we define the damage degree in three levels and collect statistics on the number of damaged objects in the building at each level.

(4) Symmetry surface extraction and missing object localization. Missing objects cannot be detected; therefore, it is necessary to use the structural information of the building to localize missing areas. First, we extract the symmetry pattern through the analysis of the spatial distribution of the remaining objects. Then, the symmetry surface is generated parametrically. Finally, we use the symmetry surface to localize the positions of missing objects.

### 3.1. Mesh Model Reconstruction

The state-of-the-art Structure from Motion and Multi-view Stereo algorithms make it easy to generate a dense point cloud from aerial imagery. The point cloud model is a simpler representation of 3D objects than volumetric and mesh models. However, it depicts objects directly by unordered discrete points, making it unsuitable for extracting quantitative data such as volume. In contrast, volumetric and mesh models represent objects with voxels and triangular faces and make it easier to extract the information needed for quantitative analysis. Thus, we reconstruct the mesh model of the Great Wall from a precise description of the object surfaces and accurate volume data.

There are many mature software programs for rebuilding 3D scenes, such as Context Capture (Acute3D/Bentley) and Pix4Dmapper (Pix4D). Get3D software [47] is a fast and free platform for the reconstruction and sharing of 3D mesh models, developed by Dashi. In this work, we use Get3D software to build the Great Wall mesh model.

### 3.2. 3D Object Segmentation

3.2.1. Segmentation from Images

Object segmentation from images remains challenging because the edges of objects are usually surrounded by more complex and chaotic backgrounds than internal regions, making edge pixels more likely to be misclassified. Additionally, the number of edge pixels is far less than that of internal pixels, causing the model to be easily dominated by internal pixels. To improve the segmentation accuracy at the object edges, we propose an edge-enhanced method for object segmentation that takes advantage of a region-based CNN and the boundary enhancement strategy (see Figure 2). The detection pipeline takes Mask R-CNN [28] as the basis and extends it with the edge enhancement module.

The Mask R-CNN is a three-stage procedure (as shown in Figure 2a). The first stage is a feature pyramid network (FPN) [23] for low- to high-level feature extraction and multilevel feature fusion. The second stage is a region proposal network (RPN) [22] for generating candidate object bounding boxes. By setting anchors of different sizes and aspect ratios, the RPN can handle objects of various sizes. The last stage consists of three branches: classification, bounding box regression, and mask prediction. The classification and bounding box regression branches predict the category and the location of the object, respectively. The mask prediction branch is a small FCN applied to each region of interest (RoI), which predicts a segmentation mask in a pixel-to-pixel manner. The inputs of the last stage are the RoIs, which are pooled based on the candidate object bounding boxes.

To improve the segmentation at object edges, we propose an edge enhancement module along with the mainstream to strengthen the learning of edge pixels in the training

process (as shown in Figure 2b). The edge enhancement module is used to extract the gradient, which contains rich edge information and indicates the feature complexity of pixels at different positions. Because the objects are connected to each other and almost identical in texture, we extract the boundary information from the label image rather than the original image. Finally, the extracted gradient is integrated with the binary cross-entropy (BCE) loss to differentially weight pixels at different positions. The extraction of the gradient and its integration with the binary cross-entropy (BCE) is given in Equation (1):

$$Loss_{mask}(y, p) = -\frac{1}{N} \sum_{i=1}^{N} W_i * (y_i * log(p_i) + (1 - y_i) * log(1 - p_i))$$

$$W = max((\mathbb{S}_{5 \times 5} \odot y), 1.0)$$

(1)

As shown in Equation (1), we add a weight $W$ to the BCE loss, where $N$ is the number of pixels, $y_i$ is the ground-truth label of the $i$th pixel, and $p_i$ is the predicted probability given by the classifier at the end of the network. For gradient extraction, we use a $5 \times 5$ Sobel filter to perform the convolution operation $\odot$ on the label image $y$. For the non-edge areas (areas far from the edge), an additional value of 1.0 is assigned to ensure that all pixels are used for training. Figure 3 shows an example of extracting the gradient. As shown, the value of the weight map $W$ decreases from the edge of the object. For edge pixels that are located at sharp edges, their weights are relatively larger than those located at flat edges (see Figure 3b). Therefore, the weight map can differentially strengthen different areas of the object, such as internal areas, flat edges, and sharp edges, thus improving the object segmentation accuracy.

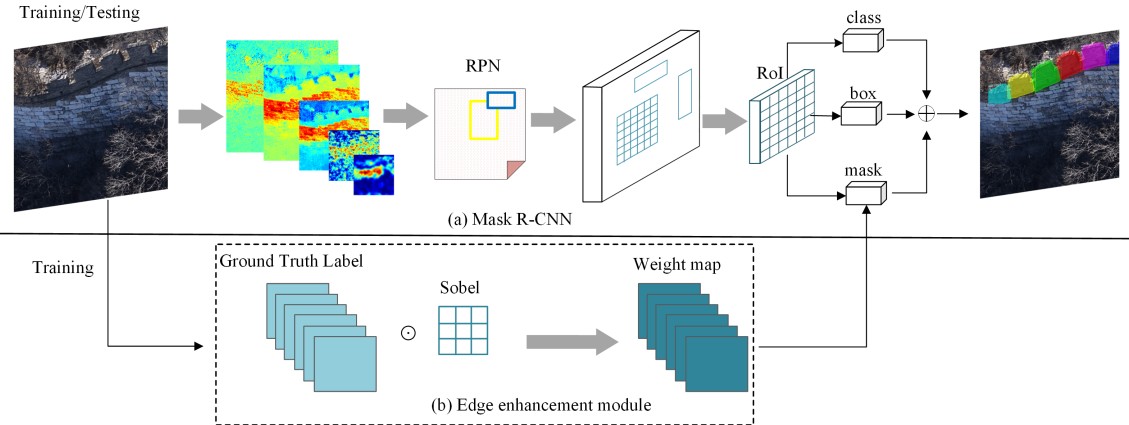

**Figure 2.** Illustration of the Edge-enhanced Mask R-CNN.

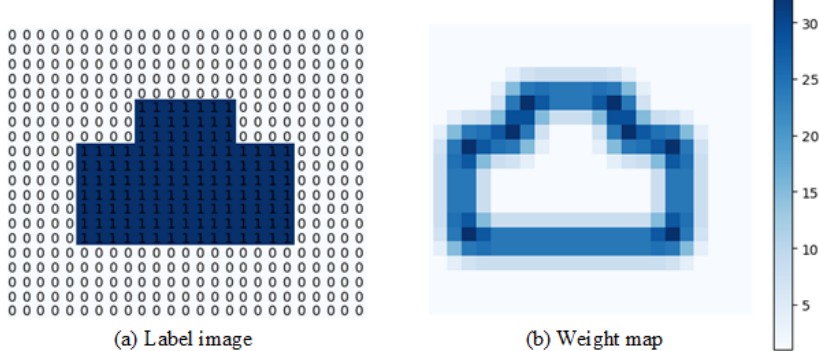

**Figure 3.** An object and its weight map. (**a**) Label image. (**b**) Weight map that differentially strengthens different areas of the object.

### 3.2.2. Segmentation from 3D Mesh Models

An image records only partial information of an object; thus, multiview image segmentation results need to be projected to 3D models and fused to build a complete object.

In the process of projecting from 2D to 3D space, the first problem that needs to be solved is eliminating background clutter, which is invisible when imaging (as shown in Figure 4). In Figure 4, we show an example of a projection process using the object of interest (the purse in the red frame in the left segmentation image). The blue points indicate the triangles of the object of interest, and the orange points indicate the background clutter. The light from the camera center passes through the foreground and background triangles at the same time; however, only the foreground triangles belong to the foreground object. To eliminate the background clutter, we adapt the ray-casting method [48] to preserve the visible triangle that is closest to the camera center among the triangles that the light passes through.

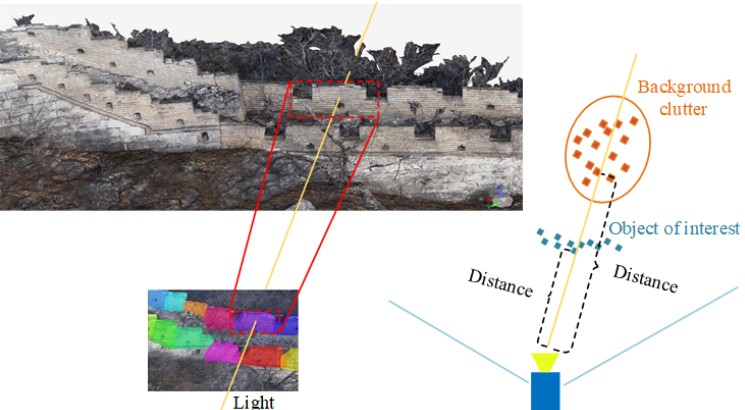

**Figure 4.** Ray-casting method for 2D to 3D projection. Each point indicates a triangle. The light passes through the foreground triangles of the purse in the red frame (in the left image) and the background clutter at the same time; however, only the triangles in the front correspond to the true object that we are interested in.

In addition to the invisible background clutter, there may still be some visible outliers coming from neighboring objects or the surrounding area, which are caused by incorrect classification in 2D object segmentation (as shown in Figure 5a). These outliers are often separated from the object and generally have fewer triangles than the object. Therefore, we simply divide all the triangles into several groups according to the characteristics of connectivity (as shown in Figure 5b). The group with the most triangles is kept as the object.

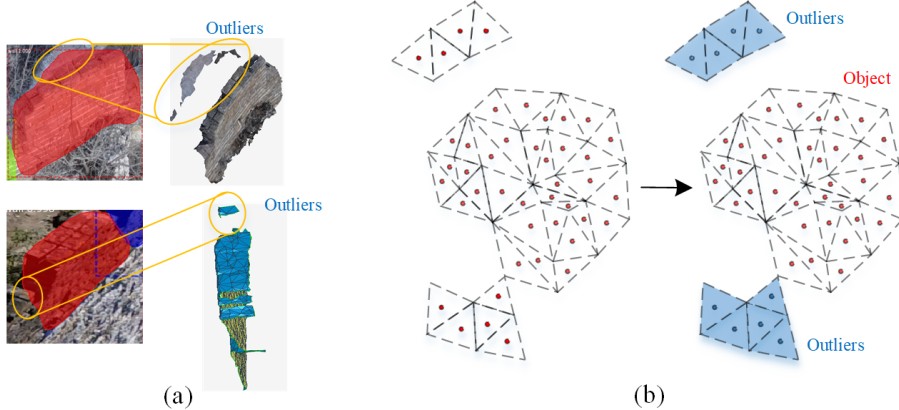

(a)  (b)

**Figure 5.** (**a**) Visible outliers that come from a neighboring object or the surrounding area. (**b**) Based on the characteristics of connectivity, outliers that have no common edge with the object are removed.

Object fragments projected from oblique images with different views are different parts of objects. To obtain the complete 3D object, we propose an integration method to integrate fragments belonging to the same object. As shown in Figure 6, integration is a two-stage procedure: integration on the same side and integration on two opposite sides. (1) First, for object fragments on the same side, we utilize the overlap of fragments to determine which fragments belong to the same object. The overlap of two fragments is defined as the ratio of common triangles to the total number of triangles in the two fragments. Two fragments are regarded as the same object and merged into one when the overlap is larger than 0.5 (as shown in Figure 6a). The integration process will not stop until two fragments cannot be merged. (2) Second, for object fragments on different sides, the common triangles mainly exist on the top of the object, causing their 3D bounding boxes to almost coincide (as shown in Figure 6b). Therefore, we generate the 3D bounding boxes and fuse the fragments with a 3D bounding box overlap larger than 0.5 into one object. The overlap of two 3D bounding boxes is defined as the ratio of the intersecting volume to the sum of their volumes. Similar to the integration of fragments on the same side, the integration process does not stop until two fragments cannot be merged.

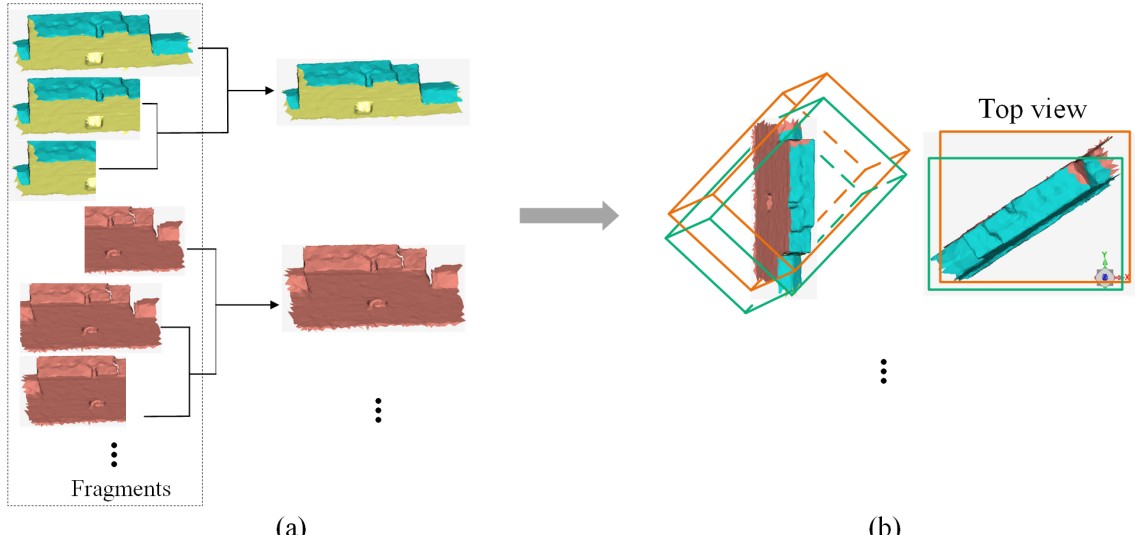

(a)                                                                (b)

**Figure 6.** Illustration of fragment integration. (**a**) Integration based on the overlap of fragments. (**b**) Integration based on the overlap of 3D bounding boxes of fragments.

### 3.3. Damage Assessment Corresponding to the Loss of Material

Further investigation on the basis of object segmentation concerns the identification of the damage level of each object. Repetitive objects tend to have the same shape, area, height, and volume; therefore, the damage condition can be generated by comparison with the undamaged object. In this work, we simply use volume reduction as the criteria of damage evaluation. To obtain the damage information of the whole structure, we define the degree of damage in three levels: (1) no or slight damage, (2) moderate damage, and (3) severe damage, and we collect statistics on the number of damaged objects of each degree. Objects with a volume reduction of less than 30% are regarded as having no or slight damage, 30 to 60% is moderate damage, and higher than 60% is considered to be severe damage. By this means, the detailed damage condition of the entire building can be generated.

As the surfaces of 3D objects extracted from the mesh model are partly enclosed, we first apply the Poisson method [49] to seal their surfaces. Then, the finite element boundary integral (FEBI) method [50] is applied to obtain the volumes of the objects. As we have no real object volume data, we manually select the undamaged object and use its volume

as the real data to calculate the volume reduction. In our work, there are two shapes of objects; thus, we select an undamaged object of each category.

### 3.4. Symmetry Surface Extraction and Missing Object Localization

It usually happens that objects in some regions are missing, causing the building to be incomplete. Locating the missing objects requires an understanding of the structure of the building. On the basis of the 3D segmentation results, the symmetry surface of the building is extracted by analyzing the spatial distribution of the 3D objects, and it is used for missing region retrieval.

#### 3.4.1. Symmetry Surface Extraction

For buildings with missing objects, the symmetry surface cannot be extracted directly. Therefore, we start from the identification of the symmetry pattern and then extract the symmetry surface parametrically. Four main steps are involved in the process of symmetry surface extraction:

1.  Grouping: Based on the spatial connectivity, these objects are divided into two groups.
2.  Facade fitting: In each group, we use a 2-degree polynomial surface equation $f_c(y,z) = ay^2 + byz + cz^2 + dy + ez + f$ to fit the facades of objects. In general, objects are perpendicular to the ground; therefore, we leave z as one of the independent variables. The other independent variable can be y or x. The surfaces are fitted using the principle of the least squares algorithm, as shown in Figure 7a.
3.  Identification of the symmetry pattern: To determine the symmetry pattern, we calculate the distance from the center of gravity of the object to the opposite facade. Due to the distances being almost equal everywhere, the symmetry pattern of this building is parallel (see Figure 7). The width of the building is the average of these distances.
4.  Symmetry surface extraction: Because the symmetry surface is located in the middle of the building, we move the objects on both sides along their normal vector directions by half the width of the building. Then, the symmetry surface is generated by fitting a 2-degree polynomial surface using these translated points, as shown in Figure 7c.

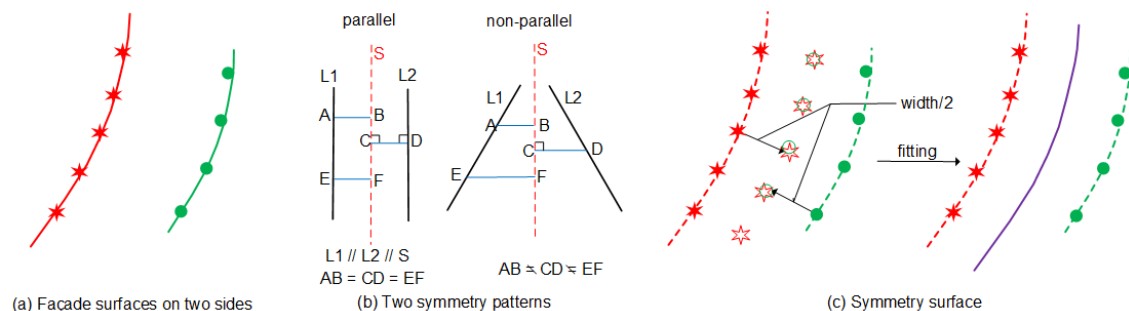

**Figure 7.** Symmetry surface extraction.

#### 3.4.2. Missing Object Localization

Symmetrical objects have a large overlap on the symmetry surface when the objects on the two sides both exist. If an object on one side is missing, the object on the other side will become a single object, without another object matching it. Taking advantage of this feature, we retrieve paired objects, single objects, and the positions of missing objects.

1.  Local axis: For each object, we project it onto the symmetry surface and generate a local axis based on its foot point on the symmetry surface. As shown in Figure 8a, the local axis takes the foot point as the origin and the normal vector v, tangent vector n, and z-axis as the three coordinate axes. The foot point is the intersection of the straight line that passes through the center of gravity of the object and is parallel to the normal vector direction of the object and the symmetry surface. The normal vector is

the partial derivative of the symmetry surface function on each axis. Then, the tangent vector is obtained as the cross-product of the normal vector and z-axis direction vector.

2. Projection overlap ratio: For each object, we transfer all the objects on the opposite side to its local axis and calculate their projection overlap ratios with the object. As shown in Figure 8b, $L_1$ and $L_2$ are the lengths of two objects, and $L_{proj}$ is the length of their projection overlap. The projection overlap ratio of the two objects is defined using $proj = \frac{L_{proj}}{min(L_1,L_2)}$.

3. Paired and single objects: An object with a projection overlap ratio greater than 0.7 with the target object will be taken as the symmetrical object. Otherwise, the target object will be recorded as a single object (as shown in Figure 8b).

4. Missing object localization: Clearly, missing objects are opposite to single objects and have an equal distance to the symmetry surface. Therefore, we extend the straight line between the centers of gravity of single objects (shown in purple) and their corresponding foot points on the symmetry surface (see Figure 8c), and then record the points with the same distances to the foot points as single objects. The coordinates of these points are the locations of the missing objects.

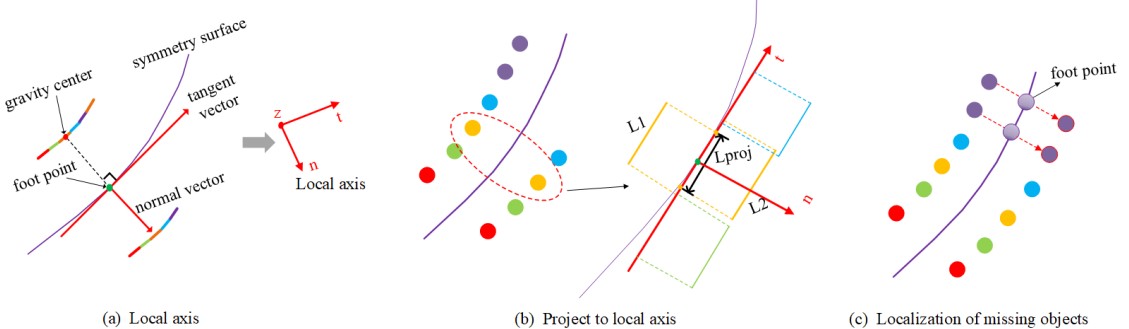

**Figure 8.** Localization of missing objects.

## 4. Experiments and Results

The experiments are divided into three parts: First, we show the segmentation performance of the proposed Edge-enhanced Mask R-CNN on both the images and the Great Wall mesh model. Second, we provide the quantitative damage evaluation results of the damaged objects and statistics on the overall damage condition of the building. Finally, we illustrate the extraction of the symmetry surface of buildings with a linear parallel symmetry structure and its application in missing object localization.

### 4.1. Datasets

#### 4.1.1. Jiankou Great Wall Oblique Images

The dataset of the Jiankou Great Wall was captured by a Falcon$^{TM}$ 8+ drone with a 75% block overlap. The Jiankou Great Wall dataset contains a total of 980 oblique images with a size of 4912 × 7360 pixels. The ground resolution range is approximately 0.5 cm to 3 cm. Because the Great Wall covers a large area, we divide it into four parts: three for training and validation and one for testing (as shown in Figure 9a,b). After eliminating the images captured from a bird's-eye view, there were 334 images for training, 90 images for validation, and 191 images for testing. The pixel size of the oblique images is too large for the CNN to process; thus, we crop the images to 1024 × 1024 pixels. Cropped images that contain Great Wall objects are selected for the experiments, which include 2773 training images, 404 valid images, and 1621 testing images. The ground truth is annotated manually (as shown in Figure 9c). Figure 9d shows the testing samples. Table 1 provides the number of images, cropped images (containing objects), and objects.

**Table 1.** The statistics of the Great Wall Dataset.

| The Number of | Images | Cropped Images (Containing Objects) | Objects |
|---|---|---|---|
| training | 334 | 2773 | 13,047 |
| valid | 90 | 404 | 2021 |
| test | 191 | 1621 | 7200 |

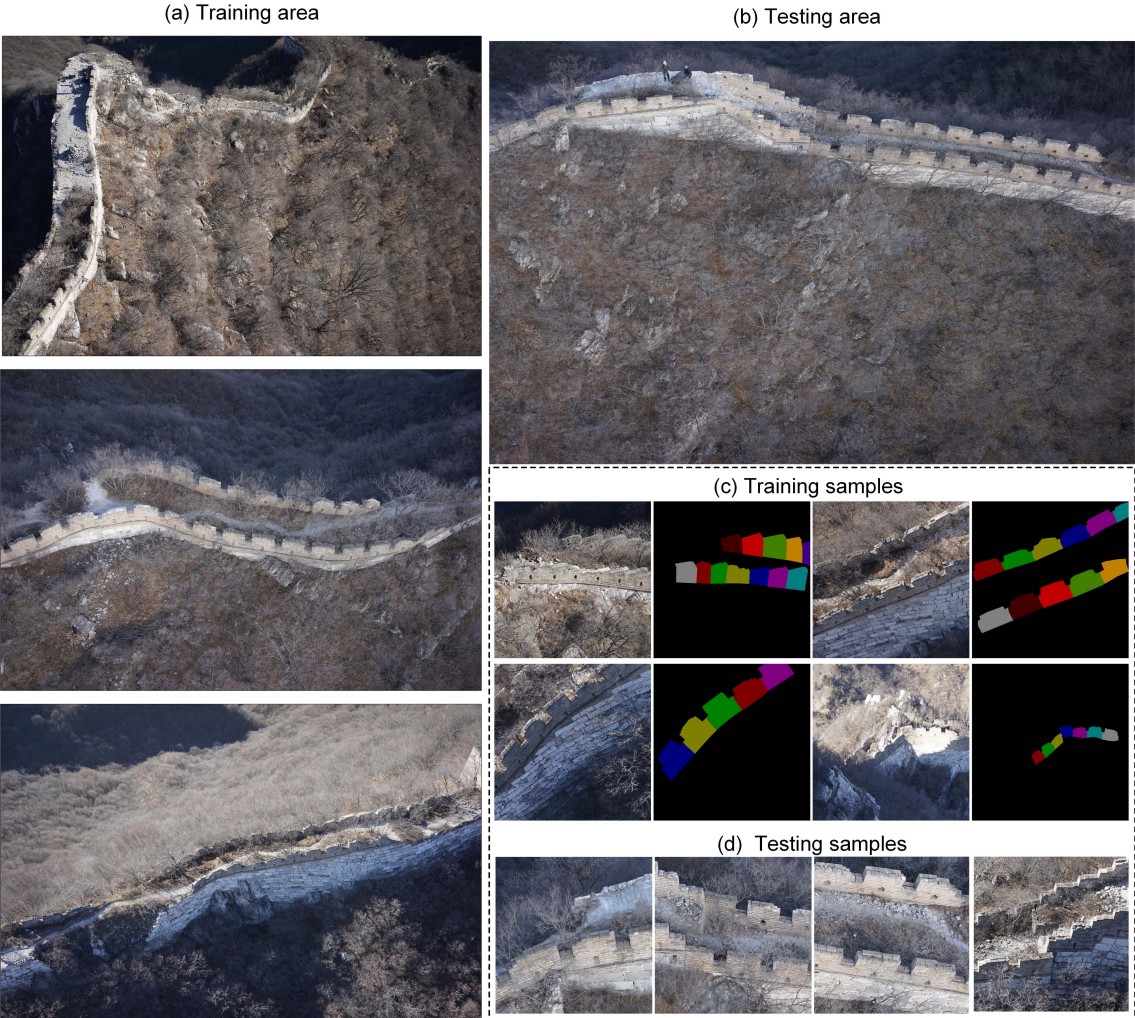

**Figure 9.** Illustration of the Jiankou Great Wall dataset.

### 4.1.2. Jiankou Great Wall Mesh Model

The Great Wall model is reconstructed using GET3D software based on 980 oblique images. As the images of the training and validation regions are used to train the CNN model, in the processes of 3D object segmentation and damage evaluation, only the test regions of the Great Wall are used, as shown in Figure 10a. This region contains 36 objects of two shape types—the curved shape and long bar shape—and seven missing objects (as shown in Figure 10b). The ground-truth (GT) objects are segmented by humans. Two undamaged objects are selected as the comparison standard (inside the red frame), which will be used to generate the volume reduction and damage degree of objects.

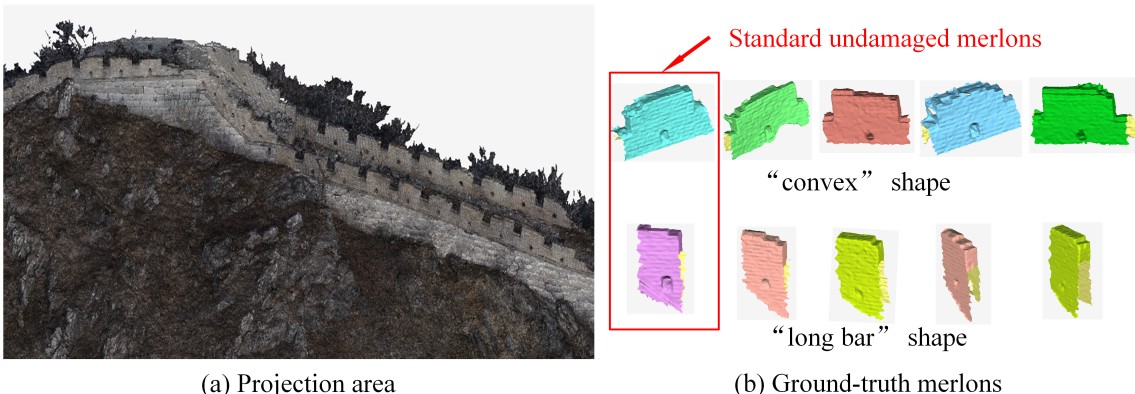

(a) Projection area  (b) Ground-truth merlons

**Figure 10.** Visualization of the projection area and ground-truth objects. (**a**) Projection area. (**b**) Two types of objects: the convex shape (first row) and long bar shape (second row). The two objects inside the red frame are selected as the standard undamaged objects.

### 4.2. Experimental Setup

We base the construction of our 2D object segmentation model on a successful Mask R-CNN, which was pretrained on the MS COCO dataset. We use ResNet101 [26] as the backbone, and the code is implemented by [51]. We train on one GPU (NVIDIA TITAN XP, 12 GB memory) for 35,300 iterations, with a starting learning rate of 0.0001. The number of images processed by each GPU is 1. We use a weight decay of 0.0001 and a momentum of 0.9.

### 4.3. Experimental Results

#### 4.3.1. 2D Object Segmentation

The results are compared in two respects: detection and segmentation. In terms of object detection, we use the commonly used criteria of average precision (AP) to assess the performance of the model. AP is usually defined as the area under the precision–recall curve, and mAP is the average of the AP. Precision represents the correctness of the predictions, while recall measures the ability of the detector to identify all positive samples. The segmented objects with an IoU larger than 0.5 with the ground-truth object are seen as positive samples, and vice versa. In terms of object segmentation, we use the IoU criteria to evaluate the segmentation precision, and the mIoU is the mean value of all objects. Table 2 shows the 2D segmentation comparison of the edge-enhanced Mask R-CNN and the original Mask R-CNN. As shown, the edge enhancement strategy improves the segmentation results by 1.61% on mIoU compared with the basic Mask R-CNN. For mAP, there is almost no improvement. Figure 11 shows the results of 2D object segmentation in oblique images, where different object instances in an image are represented by different colors. From Figure 11, we observe that the Edge-enhanced Mask R-CNN achieves better segmentation on edge pixels than Mask R-CNN.

**Table 2.** Effects of edge information enhancement on the Jiankou Great Wall test set.

| Method | Edge Information Enhancement | mAP | mIoU |
|---|:---:|---|---|
| Mask_RCNN(res101) | - | 92.90% | 82.60% |
| Mask_RCNN(res101) | √ | 93.23% | 84.21% |

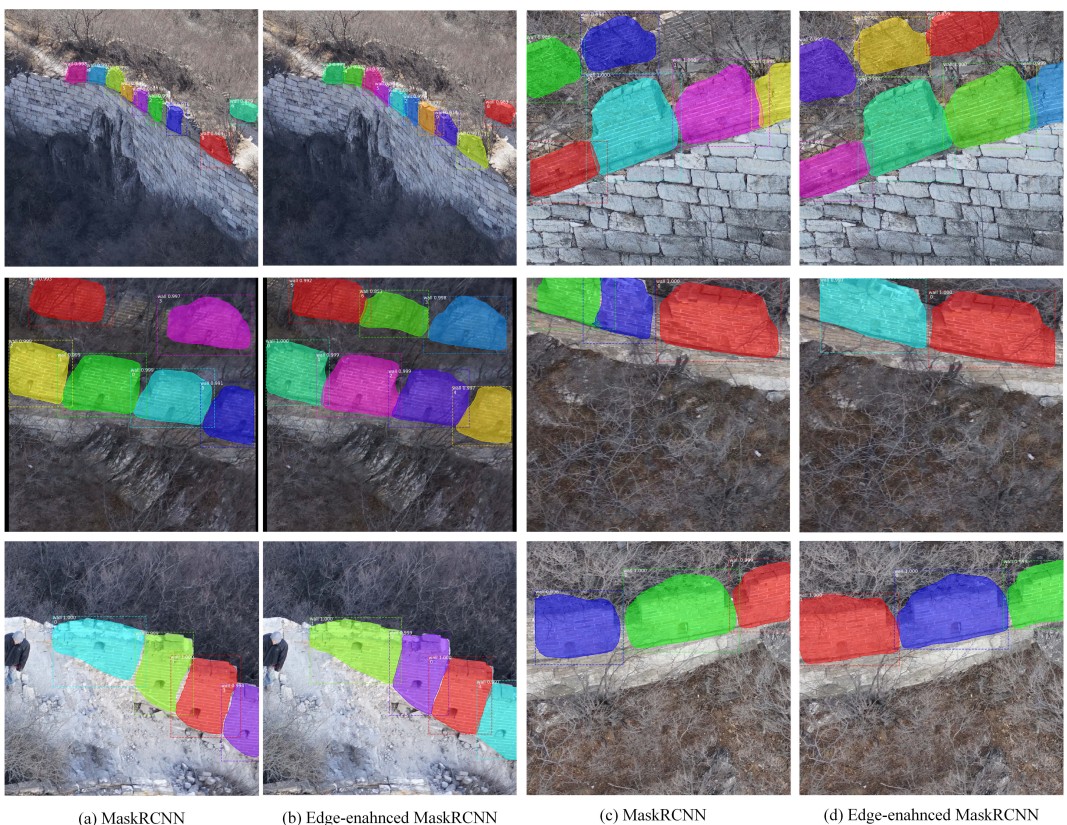

| (a) MaskRCNN | (b) Edge-enahnced MaskRCNN | (c) MaskRCNN | (d) Edge-enahnced MaskRCNN |

**Figure 11.** 2D segmentation comparison with and without the edge information enhancement strategy. Different object instances in an image are represented by different colors.

### 4.3.2. 3D Object Segmentation

Figure 12 shows the 3D segmentation comparison of our method and the ground truth. As shown in Figure 12a,b, our method achieves good performance on 3D object extraction. As segmented 3D objects do not have scores, the mAP cannot be calculated. Therefore, we use the basic criteria of precision and recall to evaluate the detection performance of the model. For segmentation, the mIoU is used. Table 3 lists the quantitative evaluation of 3D object segmentation with and without edge enhancement module. As shown, in terms of precision and recall, Edge-enhanced Mask R-CNN attains the same precision and recall as the basic Mask R-CNN. For the mIoU, the value is 2.06% higher than that of the basic Mask R-CNN method. We can see that the results of 3D object segmentation are consistent with those of 2D object segmentation.

**Table 3.** Result comparison of 3D object segmentation with and without edge enhancement module. Precision, Recall and mIoU.

| Method | Precision | Recall | mIoU |
|---|---|---|---|
| Mask R-CNN | 83.33% | 83.33% | 70.39% |
| Edge-enhanced Mask R-CNN | 83.33% | 83.33% | 72.45% |

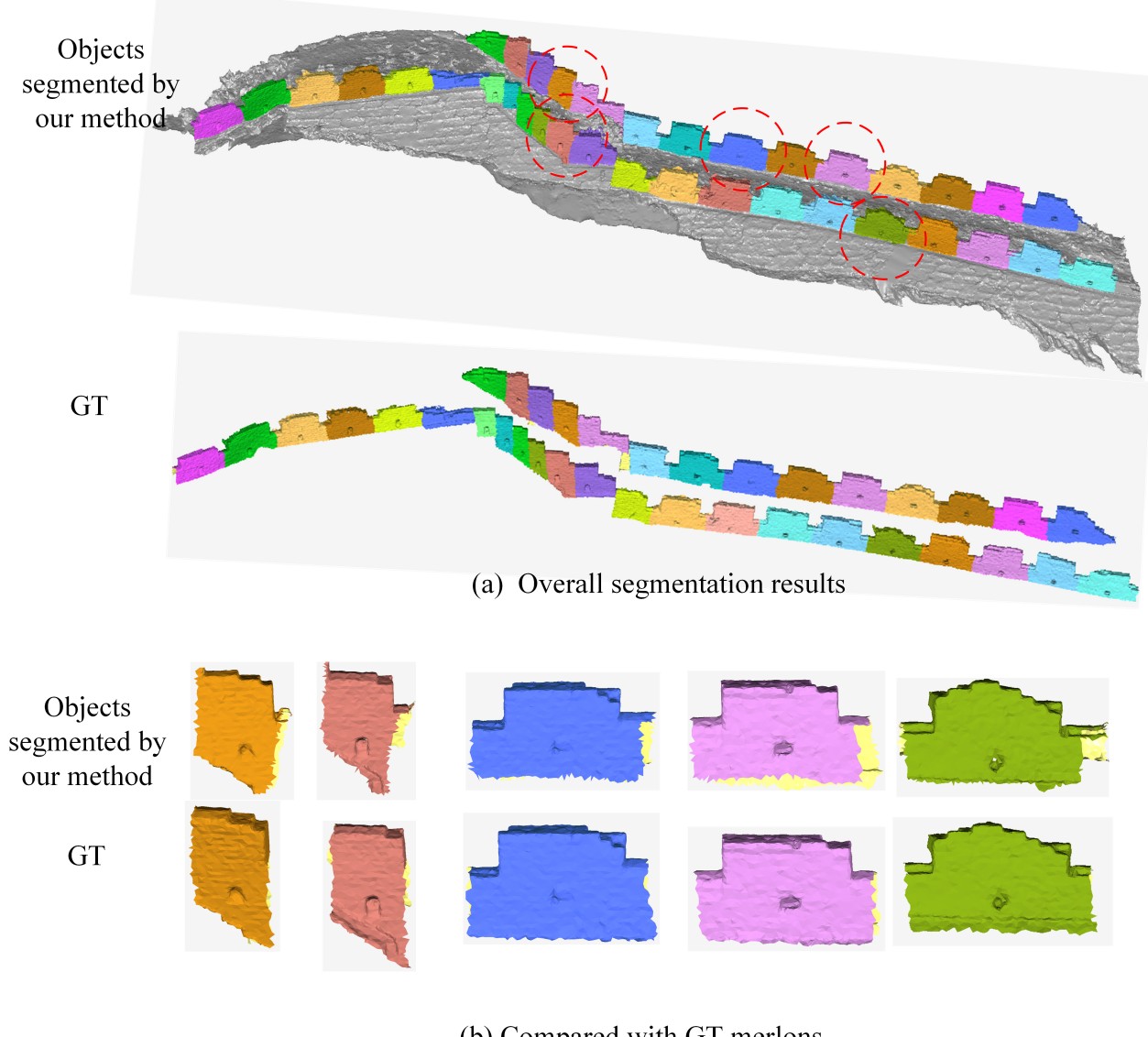

(a) Overall segmentation results

(b) Compared with GT merlons

**Figure 12.** Visualization of 3D object segmentation on the Jiankou Great Wall mesh model. (**a**) The overall segmentation results of our method and the GT objects. (**b**) The segmentation details of five objects marked with red circles in subfigure (**a**). Different object instances in an image are represented by different colors.

### 4.3.3. Damage Assessment Corresponding to the Loss of Material

Figure 13 shows the damage degree of some Great Wall objects. The first row shows the original objects segmented from the entire mesh model, which are not closed. The second row shows the objects repaired by the Poisson method. The damage degree and volume reduction of each object are given below. Table 4 collects the statistics on how many objects are damaged to each degree, from which we can learn the entire damage condition of the building. As shown, most objects are slightly damaged, with a volume reduction of less than 30%. A few objects are severely damaged, with a damage degree of 62.8%. The statistics on the damage condition can be used to guide the restoration work.

**Table 4.** The statistics of the damage degree.

| Damage Degree | Number (Total: 36) | Percent |
|---|---|---|
| No or slight damage | 33 | 91.67% |
| Moderate damage | 2 | 5.55% |
| Severe damage | 1 | 2.78% |

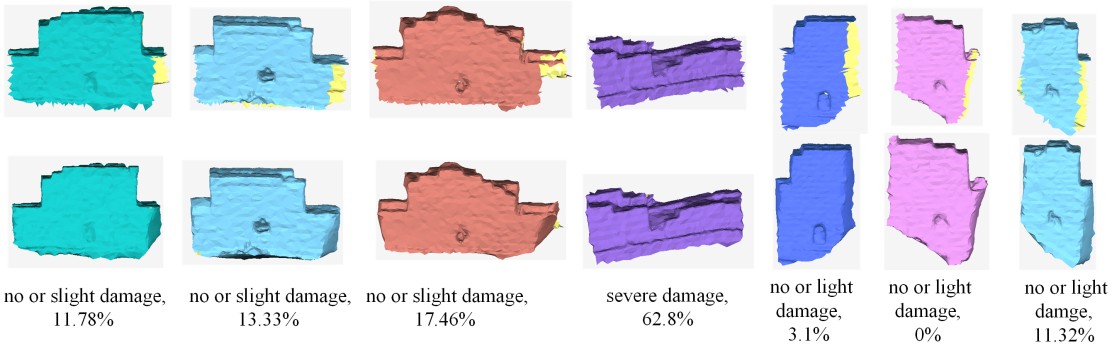

no or slight damage, 11.78%    no or slight damage, 13.33%    no or slight damage, 17.46%    severe damage, 62.8%    no or light damage, 3.1%    no or light damage, 0%    no or light damge, 11.32%

**Figure 13.** Visualization of the damage degrees of the objects. The first row shows the original objects extracted from the mesh model, which are not closed. The second row shows the objects repaired by the Poisson method. The yellow areas represent the inner surfaces of 3D objects. The damage degree and volume reduction of each object are given below.

### 4.3.4. Symmetry Surface Extraction and Missing Object Localization

Figure 14a,b shows the fitting surfaces of facades on two sides and the extracted symmetry surface, respectively. Because the distances between objects and the opposite facades are almost equal everywhere, the symmetry surface is in the middle of the two facades and parallel to them. Figure 14c shows the paired objects and single objects in 3D space. Paired objects are represented in the same color. The existence of single objects reflects the absence of missing objects. As shown in Figure 14d, the coordinate of a missing object is calculated by extending the line between the single object and its footpoint (the green point on the symmetry surface). Figure 14e,f shows the paired objects, single objects, and positions of missing objects in 2D space. To show the great significance of localizing missing objects, Figure 14g gives an example of building restoration with opposite objects. In the practical restoration process, damaged or missing objects will be repaired and completed by experts.

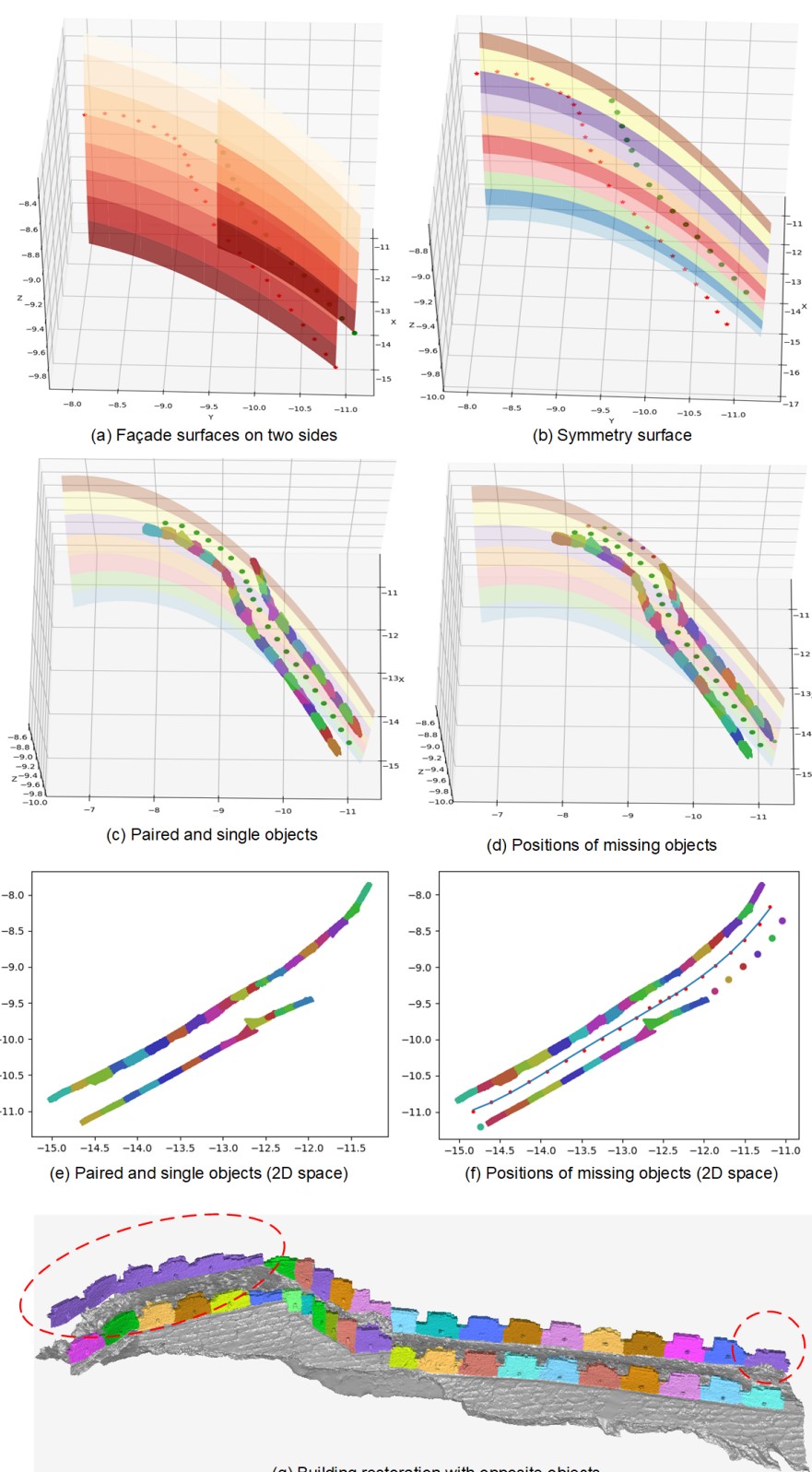

**Figure 14.** Symmetry surface extraction and missing object localization. Paired objects in (**c**) are represented in the same color.

## 5. Discussion

We propose an automatic damage estimation method for buildings with large coverage and steep terrain. The proposed method consists of three steps: segmenting objects from the

3D mesh model, quantitatively evaluating the damage condition of objects, and localizing missing objects. The contributions of the proposed algorithm are twofold: First, we propose an edge-enhanced method for more accurate object segmentation, which takes advantage of a region-based CNN and gradient enhancement strategy. Second, the symmetry surface of the building is extracted parametrically according to the analysis of the spatial distribution of the remaining objects.

Although our proposed method shows promise, there are still some limitations. (1) Concerning damage evaluation, we use the volume as the indicator to measure the damage level of an object. In our experiment, the real volumes of objects cannot be obtained on site due to the complex environment and large quantities of objects. Thus, we select undamaged objects manually and use their volumes as the standard volume for damage estimation. (2) For missing object localization, we use the symmetry surface to localize the missing objects. Therefore, our proposed method is only suitable for buildings with symmetric structures. Additionally, we do not consider the case in which objects on both sides are missing, in which case the symmetrical surface cannot be generated. (3) Only two cases of damage are considered in this work, volume reduction of a single object and missing objects in the building, which restricts the application of this method. In the future, we will conduct damage evaluations for other types of damage, such as surface cracks, deformation, and displacement.

## 6. Conclusions

In this paper, we propose a method for quantitative damage evaluation of large buildings based on drone images and CNNs. The method was tested on a case study of the Great Wall. Concerning object segmentation, the experimental results showed that the proposed method obtained a mAP of 93.23% and an mIoU of 84.21% on oblique images and an mIoU of 72.45% on the 3D mesh model. Moreover, the proposed method was effective for damaged object evaluation and missing object localization. The proposed method provides a good solution for buildings in which it is impossible to conduct damage detection on-site. In future work, we will enrich the object features to achieve higher segmentation accuracy. We will also continue to study missing object localization when objects on both sides are destroyed.

**Author Contributions:** Conceptualization, F.Z., X.H. and D.L.; Data curation, X.H.; Formal analysis, Y.G., F.Z., X.J., X.H., D.L. and Z.M.; Funding acquisition, F.Z.; Methodology, Y.G.; Writing–original draft, Y.G.; Writing–review & editing, Y.G., F.Z., X.H., D.L. and Z.M. All authors have read and agreed to the published version of the manuscript.

**Funding:** This research was funded by National Key R & D Program of China grant number 2020YFC1522703.

**Institutional Review Board Statement:** Not applicable.

**Informed Consent Statement:** Not applicable.

**Conflicts of Interest:** The authors declare no conflict of interest.

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
