# Peer review of "Deep Neural Networks for Quantitative Damage Evaluation of Building Losses Using Aerial Oblique Images: Case Study on the Great Wall (China)"

_remotesensing, doi:10.3390/rs13071321_

Round 1

Reviewer 1 Report

General considerations:

  • At the thematic level, the proposal provides a very interesting vision, as the automation of damage detection would be a very useful resource for architects and engineers in diagnostic work. Nevertheless, a thorough knowledge of the damage of a built entity is not only limited to the loss of material. This issue is an important limitation about the aspirations of the proposal, whose limitations should be assumed with more rigour and realism in the development of the argumentation of the manuscript.
  • Concerning the presentation of the contents, the document is acceptable. Nonetheless, it is recommended that authors develop proofreading to avoid common mistakes such as word confusion (e.g. in line 42: “proofs” instead of “roofs”, in line 282: “esults” instead of “results”), incorrect expressions (e.g. in line 28: “Much work utilizes”, in line 57: “and the Great Wall merlons in this paper”, or in line 184-185 “making edge pixels are easily Misclassified”), continuous repetition of the same words and expressions, incorrect use of punctuation rules (lines 96 or 154), words without a space between them (line 177), etc.
  • At a general level, the figures relating to the case study require the incorporation of graphic scales that make it possible to introduce an order of magnitude of the measurements of the elements reproduced. Unfortunately, these dimensions are not defined throughout the manuscript. Also, the analytical figures (at least figures 9,10,11,12,13 and 14) use a set of colours that seem to have a specific meaning. This meaning, if it exists, should be explained. In case the use of colours is arbitrary, this issue should also be mentioned.
  • The document contains a total of 45 employed references, of which 22 are publications produced in the last 5 years (49%), 18 in the last 5-10 years (40%), 4 than 10 years old (9%) and 1 undated (2%), implying a total percentage of 89 % recent references. In this way, the total number is sufficient, and their actuality is high.

Title, Abstract and Keywords:

  • The title, as it stands, could be improved. In this sense, it may be considered too ambitious regarding the actual contributions of the proposal. No reference is made to the heritage status of the buildings for which the research is done, nor to the type of damage (mainly loss of material) that the quantitative detection method allows. Finally, there is also no mention of the existence of a case study, which is highly valuable and representative.
  • The abstract is complete and well-structured and explains the contents of the document very well. Nonetheless, the part relating to the results could provide numerical indicators obtained in the research.

Chapter 1: Introduction

  • The first paragraph introducing the research topic (lines 27-31) gives a too simple, and even poor, view of the problems related to the diagnosis of heritage buildings and should be revised and completed with citations to authority references.
  • On a general level, the study of automatic damage detection techniques is reasonable, and the explanation of the objectives of the work may be valid. However, the limitations regarding the nature of the damage that can be detected by the method are not rigorously assumed and justified. Additionally, there are ideas, and even expressions, that are reproduced too often throughout this section, and even throughout the rest of the document.
  • Finally, a serious and illustrative introduction of the case study on which the method will be applied is missing. Explanations concerning the basic characteristics of the monument are practically non-existent throughout the document.

Chapter 2: Related work

  • It is considered that this chapter may be somewhat repetitive with the previous chapter. This issue is exemplified in the use of the references cited in section 2.1, which is very similar to those used in the previous chapter.
  • In addition, it is necessary to indicate that references to authors and works should not only be made with the number of the citation, but that hyperlinks should be accompanied by the names of the authors and works that are directly referred to in the text. This issue is also detected in later chapters of the manuscript.
  • Finally, it is necessary to point out that in section 2.3, the proposed levels of damage provided by the authors should be linked to levels referring to the degree of instability and risk that can be caused by them. It should be remembered that a building is not only a sculptural form and that they house people. To incorporate such issues in the document, situations linked to the limit states of a building should be considered, mainly the ultimate limit states and the serviceability limit states.

Chapter 3: The method

  • Based on the complexity of the contents developed in chapter 3, it is noted that the scheme in figure 1 could be even more complex and detailed in the explanation of the processes.
  • Except in section 3.1, there appears to be no indication of the computational tools and software resources used to carry out the methods presented. Similarly, the instrumental resources are specified in chapter 4, but not in chapter 3. These issues could be presented in a more orderly and clear manner.
  • In this (and later) chapters, authors are encouraged to use third and even fourth-level titles (e.g. 3.1.x or 3.1.x.x.x) for those headings that have an entity of their own. This could help to understand the structure of the manuscript's contents more easily.
  • In accordance with previous reflections, it is considered that section 3.3. should be entitled "Damage assessment corresponding to the loss of material". Such studies would be of particular interest for walls built with highly erodible material, such as earth. There are notable case studies in Southern Europe and North Africa.

Chapter 4: Experiments and results

  • In particular, figure 9c is designated as figure 9d and the caption of figure 10 does not make textual reference to situations a and b. Furthermore, line 325 mentions figure 14 when it really refers to figure 12.
  • In line 316 reference is made to the use of mAP criteria. These criteria need to be briefly explained.
  • Figure 13 does not seem to be very clear with regard to the determination of damage. What do the yellow areas mean? where does the loss of material occur in the element with a damage level of more than 60%?

Chapter 5: Discussion

  • The first paragraph of this chapter (lines 356-364) provides repetitive reflections in relation to everything explained above on the manuscript.
  • I agree that the technique actually seems more appropriate for the automatic production of heritage entities in 3D models than for the realisation of quantitative and automated damage diagnostics.
  • I recommend including the limitations regarding the consideration of damage indicated in this review in the limitations assessment (lines 376-380). This part of the document can be improved and completed with more rigour.

Chapter 6: Conclusions

  • The chapter on conclusions appears mostly as a summary.
  • After all that has been read, this technique can be considered as a complement to on-site diagnostic work, but it certainly does not seem to be a substitute for this work in its current state

Final evaluation

In summary, the research is interesting and provides valuable results, but the current document has several weaknesses that must be strengthened in order to obtain a documentary result that is equal to the value of the publication.

Reviewer 2 Report

This paper studies the damage assessment method of linear repetitive structure. The first is 2D and 3D target detection and image segmentation of repetitive structure. For this reason, the paper claims to propose improvements to Mask-RCNN to make the model more sensitive to edge information. Then the paper proposes to select a non-damaged repetitive structure as a reference, and use the ratio of the volume of the damaged part to the non-damaged repetitive structure to measure its damage. Finally, the paper proposes a method for defining the symmetry plane of the linear repetitive structure, and judges the loss of a single object in the linear repetitive structure based on the symmetry plane. The topic selection of this thesis is relatively new, and the selected objects are also representative. But the paper also has some shortcomings.

  1. The writing of this paper is cumbersome, and there are many parts that are not clearly described, making it difficult for readers to understand. For example, the acquisition process and methods of Multiview Images, eliminating background clutter, visible outliers, and Object fragments integration, also have difficulty in reading. However, the easier-to-understand parts such as symmetry plane extraction and Missing object localization in the paper are cumbersome.
  2. The descriptions in lines 206 and 262 are difficult to understand. It is recommended to add a legend or change the expression.
  3. In rows 210 and 211, what are optimized fragments, common triangles, and total triangles?
  4. In the Edge-enhanced Mask R-CNN section, the reviewer believes that the paper only adds a morphological gradient preprocessing to the input samples of Mask-RCNN, which belongs to the preprocessing of sample data, and does not make obvious to Mask-RCNN. Improve. It is difficult to become an obvious innovation point for Mask-RCNN. Besides, what is a sober filter? What's the point of using it on a ground-truth map?
  5. Is there a basis for the classification of Damage evaluation in section 3.3?
  6. The description of b in Figure 7 is cumbersome.
  7. How to realize Symmetry surface conversion in line 259 when only one parameter of "half of the building width" is known?
  8. How are the parameters described in lines 266 and 271 calculated?
  9. In the experimental part, both Damage evaluation and Symmetry surface extraction and missing object localization have no quantitative evaluation indicators, and it is difficult to explain the feasibility and advantages of the method proposed in this paper.
  10. The label in Figure 3 is incorrect, and the spelling in line 282 is incorrect.
  11. The description in line 343 should not appear in the experimental section.

Reviewer 3 Report

The authors propose a photogrammetry-based method for quantitative damage evaluation of large heritages with linear repetitive symmetry structure based on drone images. The methodology is tested on the case study of the Great Wall-China heritage.

The paper is well structured. The work is solid and generates a tool which is useful for the damage investigation of large cultural buildings. The quality of the pictures is good, as well as the experimental results obtained.

Minor comment: line 282 should read, “Experiments and results”.

Reviewer 4 Report

Dear Authors

The work presented is interesting and can be a good source for scientists and engineers. The layout is clear and the research presented is understandable and transparent. The quality of the figures and graphics is of a good standard. There are some minor editing errors in the manuscript, which should still be checked in the next stage of the application, as well as the English language.
Blibliography is comprehensive and finds references in the text. I positively review this manuscript for publication.

Regards

Round 2

Reviewer 1 Report

The authors of the manuscript have made a very thorough review of each of the points raised. The authors have addressed the requests made and have provided justified answers. Also, the quality of the manuscript has improved in its revised version.

Nevertheless, I recommend that the text be proofread one last time to avoid publication with errors. For example, I can detect an error in the title itself: I assume that "nueral" refers to the term "neural". I also recommend including the country of the selected case study in the title, in this case: "the Great Wall (China)".

In any case, I consider that the article, as it stands, is acceptable.

Reviewer 2 Report

This manuscript proposes an automatic damage estimation method for buildings with large coverage and steep terrain. The quality of the revision has a magnificent improvement and can be accepted after minor modifications. In summary, the research is quite interesting and provides valuable results, but the current document has several weaknesses that must be strengthened in order to obtain a documentary result that is equal to the value of the publication. 

Minor suggestions:

(1)The authors are strongly suggested avoiding using ".etc." at the end of sentences (Line 30).

(2)Many grammar mistakes, incorrect/Inadequate writing, etc. are presented throughout the full manuscript. the English writing needs very careful and thorough revising to make it more readable.

(3)Vision technology is emerging in the field of structural health monitoring. The authors may discuss it briefly for the integrity of the general background. Some state-of-art articles are suggested:

Haque, M., Asikuzzaman, M., Khan, I. U., Ra, I. H., Hossain, M., & Shah, S. B. H. (2020). Comparative study of IoT-based topology maintenance protocol in a wireless sensor network for structural health monitoring. Remote Sensing, 12(15), 2358.

Tang, Y., Chen, M., Lin, Y., Huang, X., Huang, K., He, Y., & Li, L. (2020). Vision-Based Three-Dimensional Reconstruction and Monitoring of Large-Scale Steel Tubular Structures. Advances in Civil Engineering, 2020: 1236021.
